# Anti-Obesity and Hypocholesterolemic Actions of Protamine-Derived Peptide RPR (Arg-Pro-Arg) and Protamine in High-Fat Diet-Induced C57BL/6J Mice

**DOI:** 10.3390/nu13082501

**Published:** 2021-07-22

**Authors:** Maihemuti Mijiti, Ryosuke Mori, Bingyu Huang, Kenichiro Tsukamoto, Keisuke Kiriyama, Keita Sutoh, Satoshi Nagaoka

**Affiliations:** 1Department of Applied Life Science, Faculty of Applied Biological Sciences, Gifu University, 1-1 Yanagido, Gifu 501-1193, Japan; y6103003@edu.gifu-u.ac.jp (M.M.); z4521084@edu.gifu-u.ac.jp (R.M.); huangbingyu0609@gmail.com (B.H.); ke021216@gmail.com (K.T.); 2Fordays Co., Ltd., Tokyo 103-0016, Japan; k.kiriyama@fordays.jp (K.K.); k.sutoh@fordays.jp (K.S.)

**Keywords:** RPR, protamine, peptide, obesity, fat, cholesterol

## Abstract

Dietary protamine can ameliorate hyperlipidemia; however, the protamine-derived active peptide and its hypolipidemic mechanism of action are unclear. Here, we report the discovery of a novel anti-obesity and hypocholesterolemic peptide, RPR (Arg-Pro-Arg), derived from protamine in mice fed a high-fat diet for 50 days. Serum cholesterol levels were significantly lower in the protamine and RPR groups than in the control group. White adipose tissue weight was significantly decreased in the protamine and RPR groups. The fecal excretion of cholesterol and bile acid was significantly higher in the protamine and RPR groups than in the control group. We also observed a significant decrease in the expression of hepatic SCD1, SREBP1, and adipocyte FAS mRNA, and significantly increased expression of hepatic PPARα and adipocyte PPARγ1 mRNA in the protamine group. These findings demonstrate that the anti-obesity effects of protamine are linked to the upregulation of adipocyte PPARγ1 and hepatic PPARα and the downregulation of hepatic SCD1 via SREBP1 and adipocyte FAS. RPR derived from protamine has a crucial role in the anti-obesity action of protamine by evaluating the effective dose of adipose tissue weight loss.

## 1. Introduction

Endocrine disorders, environmental factors and genetic susceptibility cause obesity, which is a severe metabolic disease. It is typically related to other metabolic diseases such as cardiovascular disease, type 2 diabetes, hypertension, and cancer. It is characterized by an abnormal fat-accumulation in the adipose tissue that results from excessive calorie intake derived from an imbalance between energy intake and energy consumption [1,2,3,4]. Numerous nutritional intervention studies have reported that dietary proteins or hydrolysates have been shown to reduce body weight and fat mass and lower levels of cholesterol and triglycerides in plasma. For example, soybean protein is reported to effectively lower body weight and fat mass in overweight and obese subjects [5,6]. Dietary fish protein hydrolysates have been reported to affect serum lipids and postprandial glucose regulation in obese Zucker fa/fa rats [7]. Moreover, yellow catfish protein hydrolysate exhibited anti-obesity effects in mice fed a high-fat diet (HFD), compared with the simvastatin treated mice [8].

Protamine is a basic protein obtained from fish (mainly salmon) generally and consists of heterogeneous proteins which have an average molecular weight of 4500 Da. Approximately 70% of the amino acid composition of protamine is arginine [9]. Jobgen et al. reported that arginine, the main amino acid constituent of protamine, also has an anti-obesity effect [10]. Fish milt derived protamine has a low molecular weight (4–5 kDa) and protects from DNA damage. An in vivo experiment found that oral administration of protamine to rats decreased the plasma triglyceride concentration [11] and increased fecal triglyceride concentration [12]. These effects show that protamine is a strong candidate for inhibiting lipid adsorption [11,12]. Protamine exhibits both hypocholesterolemic and anti-obesity effects in rats fed a HFD [13,14,15]. However, the active peptide and molecular mechanism of protamine’s anti-obesity and hypocholesterolemic actions have not yet been elucidated. In addition, protamine hydrolysates and protamine-derived peptides, especially their bioactive peptides, have gained little attention from researchers.

Thus, we aimed to identify the active peptide derived from protamine that was related to both anti-obesity and hypocholesterolemic actions induced by protamine. Based on our research, we hypothesized that RPR derived from protamine is an active peptide related to both anti-obesity and hypocholesterolemic actions induced by protamine. To test this hypothesis, we attempted to identify novel anti-obesity and hypocholesterolemic peptides derived from protamine in animal studies. We also investigated the effects of the novel anti-obesity and hypocholesterolemic peptide, Arg-Pro-Arg (RPR) derived from protamine, in mice fed a HFD.

## 2. Materials and Methods

### 2.1. In Silico Digestion

In this research, protamine derived from chum salmon milt (Oncorhynchus keta) was used. Chum salmon-derived protamine was digested using in silico analysis with trypsin by PeptideCutter (https://web.expasy.org/peptide_cutter/, accessed on 1 March 2019). Protamine is composed of 33 amino acids (MPRRRRSSSRPVRRRRRPRVSRRRRRRGGRRRR), and arginine accounts for approximately 70% of the constituent amino acids [16].

### 2.2. Animals and Diets

Six-week-old male C57BL/6J mice were purchased from Japan SLC (Hamamatsu, Japan). Mice were housed individually in standard plastic rodent cages and placed in a room that was kept at 22 ± 2.0 °C; there was a 12-h cycle of light (8:00–20:00) and dark, and food and water were freely available. All mice were fed a commercial non-purified diet (MF, Oriental Yeast, Osaka, Japan) and tap water ad libitum for 3 or 5 days before their division into three weight-matched groups. Subsequently, each group was given free access to drinking water and fed the experimental diets for 50 days.

There are three experimental groups as shown in Table 1.

The composition of the experimental diets (HFD) was based on the formula recommended by the American Institute of Nutrition (AIN-93G) [17] and contained, in weight percent, casein, 25; soybean oil, 4.0; lard, 30; cellulose, 6.5; AIN93 mineral mixture, 3.5; AIN93 vitamin mixture, 1.0; choline chloride, 0.4; sucrose, 9.87; and corn starch, 19.73. RPR or protamine was added to the diet at a nitrogen level equivalent to that of a casein diet at the expense of carbohydrates. Food consumption and body weight of the mice were measured everyday during the feeding period. At the end of the experimental period, the mice were starvated for 22 h, and sacrificed under isoflurane anesthesia by drawing blood from the heart. The liver and visceral fat pads were then removed and weighed. The plasma, liver, and visceral fat pad samples were collected and stored at −80 °C until analysis. Fecal collection (days 46–50) was used to determine the fecal lipid content. The Ethics Committee on Animal Experiments at Gifu University approved all experimental protocols (permit number: 2019-022). All experiments used in this study were performed according to the experimental guidelines and regulations of Gifu University.

### 2.3. Biochemical Analyses

Lipid concentrations in serum, liver, and feces were measured using commercially available kits as follows: Triglyceride E-test Wako (Wako Pure Chemical, Osaka, Japan) used for serum, liver, and fecal triglyceride, Cholesterol E-test Wako (Wako Pure Chemical, Osaka, Japan) used for serum, liver, and fecal cholesterol, and with HDL-Cholesterol E-test Wako (Wako Pure Chemical, Osaka, Japan) used for serum HDL-cholesterol, Phospholipid C-test Wako (Wako Pure Chemical, Osaka, Japan) used for liver, and fecal phospholipid, and TBA-test Wako (Wako Pure Chemical, Osaka, Japan) used for fecal bile acid. Liver and fecal lipids were extracted by the method of Folch et al. [18], and total lipids were measured gravimetrically by the method of Nagaoka et al. [19].

### 2.4. RNA Preparation from Mouse Tissues and Real-Time PCR

A total 5500 ng RNA isolated from mouse liver and white adipose tissue (WAT) was reverse-transcribed to cDNA using a High-Capacity cDNA Archive Kit (Applied Biosystems, Foster City, CA, USA). Real-time PCR was performed using a StepOnePlus^TM^ Real-time PCR system (Applied Biosystems) and SYBR^®^ Premix Ex Taq (Takara, Shiga, Japan), according to the manufacturer’s protocol. The amplification protocol was as follows; the temperature profile for the reaction was an initial denaturation stage of 95 °C at 30 s, then a two-step program was developed for 40 cycles including denaturation at 95 °C for 5 s, an annealing step and extension step at 60 °C for 30 s, respectively. The following primers were used in the assay based on the NCBI databases (Table 2). Relative gene expression was calculated using the 2^−ΔΔCt^ method.

### 2.5. Statistical Analyses

Values are expressed as means  ±  SEM. The statistically significant differences were evaluated using one-way ANOVA and Dunnett’s tests [20]. Differences were considered significant at * *p* < 0.05, ** *p* < 0.01. The statistical analysis was performed by JSTAT ver 8.2.

## 3. Results

### 3.1. In Silico Analysis

Protamine was digested with trypsin using PeptideCutter. Protamine tryptic hydrolysate consists of MPR, RRR, SSSRPVR, RRRRPR, VSR, RRRRR, GGR, and RRR by PeptideCutter (protamine is not hydrolyzed by pepsin) in silico. Among them, RRRRPR is hydrolyzed one by one of arginine from the N-terminus by trypsin [21] and finally produces RPR (Figure 1) in silico. Thus, RPR cannot be hydrolyzed by trypsin and pepsin.

### 3.2. Body Weights, Food Intake, Liver Weights

We examined body weight gain, food intake, and liver weight of each mouse in the experiment. There were no significant differences between the control group and protamine or RPR groups regarding growth parameters, namely, initial body weight, food intake, and liver weight. Body weight gain in the protamine group was significantly lower than that in the control group (Figure 2a–c). There were no significant differences in the food intake on each day among the groups (date is not shown). The average daily intake of protamine or RPR is 4.71 ± 0.17 g/kg body weight or 0.5 ± 0.02 g/kg body weight respectively for 50 days.

### 3.3. WAT Weights

To investigate whether RPR derived from protamine or protamine induced anti-obesity action in mice fed an HFD diet, various WAT weights were measured. The perirenal or subcutaneous WAT weight was significantly lower in the protamine and the RPR groups than in the control group, and epididymal WAT weight decreased in the RPR group compared to the control group. Total WAT weights decreased significantly in the protamine and RPR groups compared to the control group (Figure 3a,b), suggesting that protamine and RPR suppressed body weight gain by decreasing total WAT weights.

### 3.4. Serum and Liver Lipid Parameters

As shown in Figure 4a, serum total cholesterol and HDL-cholesterol in the protamine group or RPR group was significantly lower than that in the control group. Serum non-HDL-cholesterol level and atherogenic index was significantly decreased in the protamine group compared to control group (Figure 4a,b). Serum triglyceride levels were not significantly different among the groups (Figure 4c). Liver total lipid, phospholipid, triglyceride, and cholesterol levels were significantly unchanged among the groups (Figure 4d).

### 3.5. Fecal Weights and Fecal Lipid Parameters

We evaluated the effect of dietary RPR or protamine on fecal lipid excretion in mice. Fecal weights were not significantly different among the groups (Figure 5a). The fecal excretion of cholesterol was significantly higher in mice treated with RPR or protamine than that of the control (Figure 5c), and fecal bile acid excretion was significantly increased in mice treated with RPR or protamine when compared with the control (Figure 5c). The fecal total lipid, phospholipid, and triglyceride levels were not significantly different among the groups (Figure 5b,c).

### 3.6. Effect of RPR or Protamine on the mRNA Levels of Lipid Metabolism-Related Genes in the Liver of Mice

We determined the mRNA expression levels of lipid metabolism-related genes in the liver to evaluate the effects of protamine and RPR using real-time PCR (Figure 6). Hepatic SCD1 or SREBP1 mRNA levels were significantly decreased in the protamine group compared to those in the control group. Hepatic PPARα mRNA levels were significantly higher in the protamine group than in the control group. We evaluated the expression of sterol regulatory element-binding protein 2 (SREBP2) related to HMG-CoA reductase (HMGCR) and LDL receptor (LDLR) gene expression. LDLR expression was significantly upregulated in mice fed the RPR diet compared to that in the control group. When protamine was added to the HFD, SREBP2, HMGCR, and LDLR expression levels were upregulated compared to those in the control group (Figure 6). There were no significant differences between the control group and protamine or RPR groups in the other mRNA levels (Figure 6).

### 3.7. Effect of RPR or Protamine on the mRNA Levels of Lipid Metabolism-Related Genes in Epididymal WAT of Mice

We determined the mRNA expression levels of lipid metabolism-related genes in epididymal WAT to evaluate the effect of protamine and RPR using real-time PCR (Figure 7). PPARγ1 mRNA levels were significantly increased by feeding protamine compared to those in the control group. FAS mRNA levels in epididymal WAT were significantly lower in the protamine group than in the control group (Figure 7). SCD1, ACOX, AMPK, PPARα, PPARγ2, CPT1, SREBP1, adiponectin, and HSL mRNA levels were not significantly different among the groups.

## 4. Discussion

This is the first research to identify RPR (Arg-Pro-Arg) as a novel anti-obesity and hypocholesterolemic tripeptide derived from protamine in mice. The degree of anti-obesity action of RPR treatment was almost identical to that of the protamine treatment, as measured by the WAT weight (Figure 3). Thus, RPR derived from protamine has a crucial role in the anti-obesity action of protamine by evaluating the effective dose (protamine: 4.71 ± 0.17 g/kg body weight, RPR: 0.5 ± 0.02 g/kg body weight) of adipose tissue weight loss. To our knowledge, there are no reports of both anti-obesity and hypocholesterolemic actions by a tripeptide-like RPR derived from any protein origin. Thus, RPR is the only tripeptide that has both anti-obesity and hypocholesterolemic functions. The effect of RPR or protamine on visceral adipose tissue weight loss is particularly relevant, because visceral fat accumulation is the major risk factor for the progress of chronic degenerative diseases such as type 2 diabetes due to glucose intolerance and hyperinsulinemia resulting from the resistance of insulin [22]. Serum HDL-cholesterol level was significantly decreased in the protamine group and RPR group and non-HDL-cholesterol was significantly decreased in the protamine group (Figure 4a). It has been reported that total cholesterol, HDL-cholesterol, and LDL-cholesterol were significantly decreased in rats administrated protamine for 7 weeks [15]. These results resemble our study. Furthermore, we will investigate the level of free fatty acid to clarify protamine effects in future studies. Plasma insulin and glucose levels are also important factors in obesity and hypocholesterolemia. Therefore, we need to measure the insulin and glucose levels in mice fed RPR or protamine in future studies.

Decreased circulating total cholesterol and LDL cholesterol concentrations are associated with a reduction in the risk of cardiovascular disease [23]; therefore, it is interesting to identify nutrients with hypocholesterolemic functions that may support the prevention of cardiovascular disease. Mice fed a RPR or protamine diet had a lower serum cholesterol concentration than the control mice. The fecal excretion of cholesterol was significantly higher in the protamine or RPR groups than in the control group, and fecal bile acid excretion significantly increased in the protamine and RPR-treated animals when compared with the controls (Figure 5c). These findings suggest that the hypocholesterolemic action of RPR or protamine is related to the inhibition of intestinal cholesterol absorption and was measured by the increase in fecal steroid excretion, similar to previous findings [14,19]. Regarding cholesterol metabolism, we evaluated the mRNA levels of SREBP2, HMGCR, and LDLR. LDLR mRNA levels were significantly increased in mice fed a RPR diet compared to those in the control group. When protamine was added to the HFD, SREBP2, HMGCR, and LDLR, mRNA levels were upregulated compared to those in the control group (Figure 6). These findings resemble those of an earlier study which have demonstrated that dietary addition of saponin has increased the uptake of LDL-cholesterol in hepatocytes even though cholesterol synthesis has not been downregulated, which may explain the reduction in plasma LDL-cholesterol [24]. Our study suggests that the serum cholesterol-lowering action of RPR or protamine is induced by LDLR activation.

Hepatic SCD1 or SREBP1 mRNA levels were significantly decreased in the protamine group compared to those in the control group (Figure 6). Some studies have indicated that SREBPs co-regulate cholesterol and fatty acids and that SREBP1 is a key regulator that activates the expression of lipogenic genes in response to hepatic insulin signaling [25]. SREBP1 is a transcription factor for controlling *de novo* lipogenesis genes, for example, FAS [26] or SCD1 [27]. A previous study suggested that the enhanced triglyceride synthesis was strongly dependent upon the expression of SCD1 gene and that *SCD1*-KO mice showed the reduction of fatty acid and triglyceride synthesis by the high-carbohydrate feeding [27]. *SCD1* ablation significantly decreased SREBP1 mRNA level, which was consistent with the observation that *SCD1*-KO mice showed a reduced expression and maturation of SREBP1, which was specifically rescued by oleate treatment. Oleate can modify the SREBP1 protein and enhance its nuclear translocation, which increases the expression of lipogenic genes [28]. Thus, the suppression of hepatic SCD1, via SREBP1 related to lipogenic genes by protamine feeding, is responsible for the reduced WAT weight. The decrease in body weight gain in mice by protamine feeding may be attributed to the increased energy expenditure and oxygen consumption by the enhancement of fatty acid oxidation and thermogenesis in the liver, muscle, and brown adipose tissue, similar to the effects of SCD1 deficiency in *SCD1*-KO mice [29]. To confirm our hypothesis, we will measure the brown adipose tissues, or check the basal metabolic rate of mice fed RPR or protamine in future study.

The study of Lodhi et al. has shown that the specific deletion of FAS in adipose tissue increases brown fat-like adipocytes in WAT and increases energy expenditure, which ameliorates diet-induced obesity [30]. Thus, as the adipocyte FAS mRNA levels were significantly decreased in the protamine group compared to those in the control group, the suppressed expression of FAS induced by protamine might be related to the suppression of lipid accumulation in the WAT (Figure 3b). Peroxisome proliferator-activated receptors (PPARs) are members of the nuclear receptor superfamily and have three distinct isoforms, α, γ, and δ, whose ligand specificities and tissue distributions are different [31]. PPARα is primarily expressed in the brown adipose tissue and liver, while the expression of PPARγ is abundant in the adipose tissue and works as an important transcription factor in adipocyte differentiation [32]. Thus, PPARγ agonists have been applied to control insulin resistance induced by obesity [33]. Moreover, PPARα functions as the physiological stimuli in the fasting state, and it is hypothesized that the free fatty acids transported from adipose tissue to the liver act as ligands and activate PPARα [34]. Therefore, the significant increase in hepatic PPARα mRNA levels observed in the protamine group compared to the control group (Figure 6) might be linked to the reduction of lipid accumulation in the WAT (Figure 3b). These results suggest that dietary protamine decreases WAT weight by reducing hepatic SREBP1 and SCD1 mRNA levels and activating adipocyte PPARγ1 and hepatic PPARα mRNA levels. Thus, we can expect protamine and RPR derived from protamine to be useful against both hypercholesterolemia and obesity-induced metabolic disorders.

The molecular mechanism related to the suppression of SCD1 via SREBP1 or the activation of adipocyte PPARγ1 and hepatic PPARα by RPR or protamine is unclear. Two relevant mechanisms have already been discovered. Lactostatin (IIAEK), an in vivo active novel hypocholesterolemic pentapeptide, which mediates the calcium-channel-related mitogen-activated protein kinase (MAPK) signaling pathway for cholesterol degradation in HepG2 cells [35], and the novel intestinal alkaline phosphatase (IAP)-dependent intestinal ATP-binding cassette transporter A1 (ABCA1) signaling pathway for intestinal cholesterol absorption [36]. We have also previously discovered that phenylalanine-proline (FP) has in vivo hypocholesterolemic action in rats [37]. Interestingly, as FP-induced cholesterol-lowering effect disappeared in *PepT1*KO mice, the cholesterol-lowering effect of FP was mediated through PepT1 in mice [37]. These observations have suggested that PepT1 is an important target for improving cholesterol metabolism [37]. As RPR may enter hepatocytes after intestinal absorption via PepT1 [38], RPR may function in hepatocytes in vivo. The possibility that RPR could mediate the novel regulation of lipid metabolism related to the MAPK pathway, IAP pathway, or PepT1 is currently being investigated.

Taken together, this study shows that RPR derived from protamine is a new biological agent for the creation of functional foods and medicines that are expected to prevent and improve obesity and hypercholesterolemia.

## 5. Conclusions

In summary, our results suggest that RPR derived from protamine or protamine supplementation ameliorates hypercholesterolemia and obesity in HFD mice (Figure 8). RPR or protamine feeding exhibited a hypocholesterolemic effect via fecal cholesterol and bile acid excretion. This may be due to increased expression of LDLR associated with upregulated hepatic LDLR mRNA levels. Protamine significantly suppressed the mRNA expression of hepatic SCD1 through SREBP1 and adipocyte FAS. Protamine upregulated adipocyte PPARγ1 mRNA and hepatic PPARα mRNA levels. The changes in protein expression caused by these mRNA levels may have affected the suppression of WAT weight. RPR derived from protamine has a crucial role in the anti-obesity action of protamine by evaluating the effective dose of adipose tissue weight loss. However, the mechanism of the hypolipidemic action of dietary RPR may slightly differ from that of protamine because of the differences in the effects on gene expression of lipid metabolism. These findings show that dietary RPR derived from protamine and protamine can prevent or treat obesity.

## Figures and Tables

**Figure 1 nutrients-13-02501-f001:**
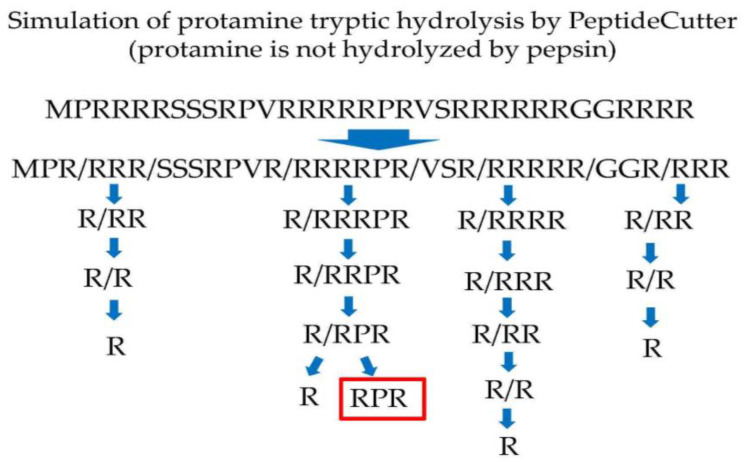
Simulation of protamine tryptic hydrolysis by PeptideCutter (https://web.expasy.org/peptide_cutter/, accessed on 1 March 2019) in silico.

**Figure 2 nutrients-13-02501-f002:**
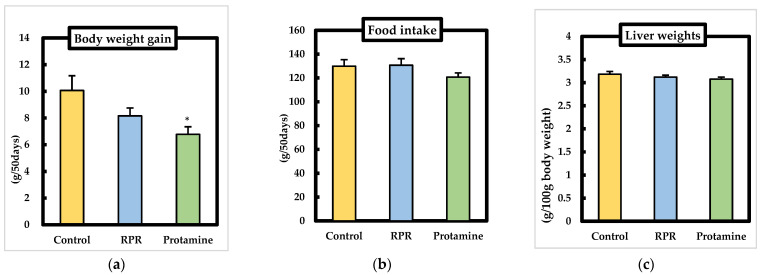
Effect of RPR or protamine on body weight gain, food intake, and liver weight in C57BL/6J mice with high-fat diet-induced obesity. (**a**) The body weight gain, (**b**) food intake, and (**c**) liver weights were measured after 50 days of high-fat diet treatments in mice. Means ± SEM (*n* = 8). Differences from the control group were calculated using Dunnett’s test (* *p* < 0.05).

**Figure 3 nutrients-13-02501-f003:**
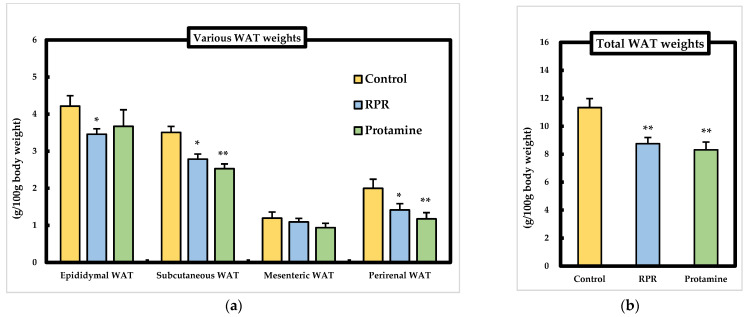
Effect of RPR or protamine on adipose tissue weights in C57BL/6J mice with high-fat diet-induced obesity. (**a**) The epididymal white adipose tissue (WAT), subcutaneous WAT, mesenteric WAT, and perirenal WAT, (**b**) total WAT weights were measured after 50 days of high-fat diet treatments in mice. Means ± SEM (*n* = 8). Differences from the control were calculated using Dunnett’s test (* *p* < 0.05, ** *p* < 0.01).

**Figure 4 nutrients-13-02501-f004:**
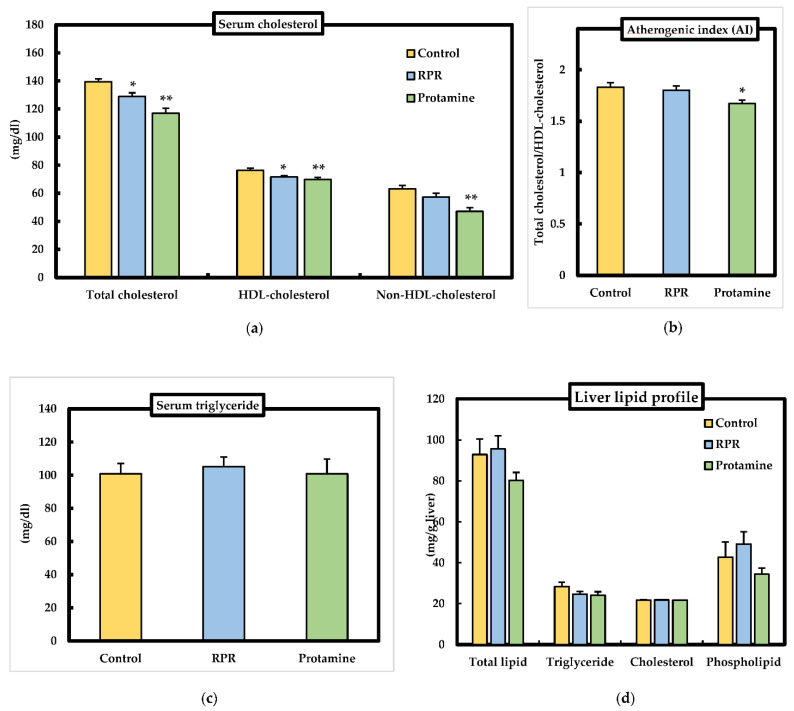
Effect of RPR or protamine on serum and liver lipids in C57BL/6J mice with high-fat diet-induced obesity. (**a**) Serum cholesterol, (**b**) Atherogenic index, (**c**) serum triglyceride, (**d**) liver lipid profiles were measured after 50 days of high-fat diet treatments in mice. Means ± SEM (*n* = 8). Differences from the control group were calculated using Dunnett’s test (* *p* < 0.05, ** *p* < 0.01).

**Figure 5 nutrients-13-02501-f005:**
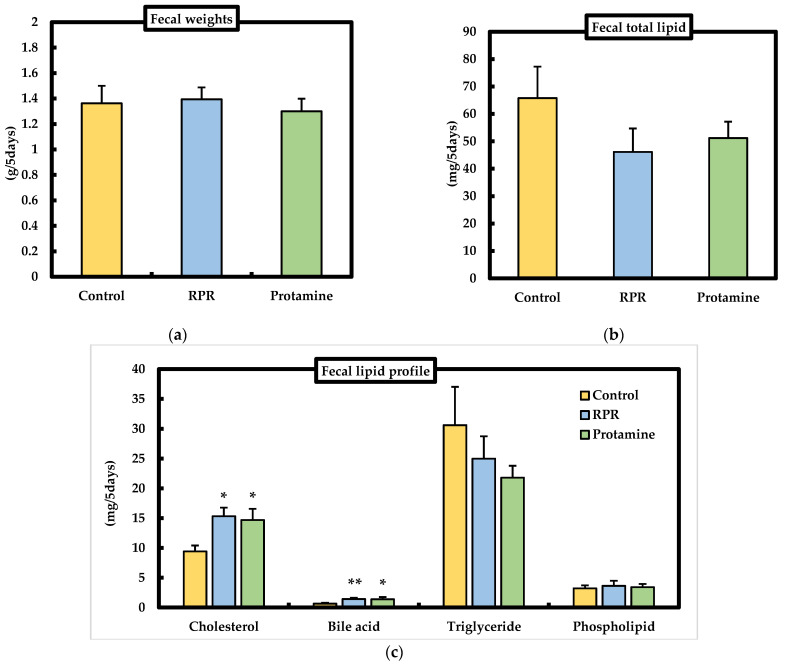
Effect of RPR or protamine on lipid content in the feces of C57BL/6J mice with high-fat diet-induced obesity. (**a**) Fecal weights, (**b**) fecal total lipid, (**c**) fecal cholesterol, bile acid, triglyceride, and phospholipid. The feces were collected on days 46 to 50. Means ± SEM (*n* = 8). Differences from the control were calculated using Dunnett’s test (* *p* < 0.05, ** *p* < 0.01).

**Figure 6 nutrients-13-02501-f006:**
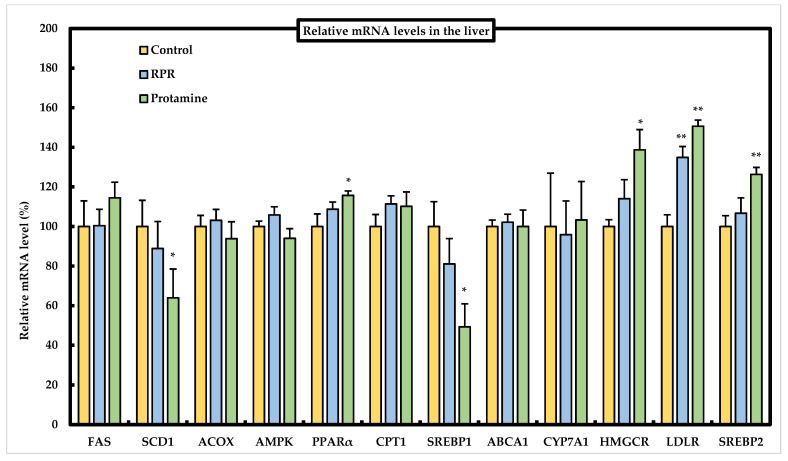
Effect of RPR or protamine on relative mRNA levels in the liver of C57BL/6J mice with high-fat diet-induced obesity. The relative mRNA levels in the liver were measured after 50 days of high-fat diet treatments in mice. The mRNA levels of the various genes in the control group are considered to be 100%. Means ± SEM (*n* = 8). Differences from the control were calculated using Dunnett’s test (* *p* < 0.05, ** *p* < 0.01). The relative mRNA levels were measured by real-time PCR.

**Figure 7 nutrients-13-02501-f007:**
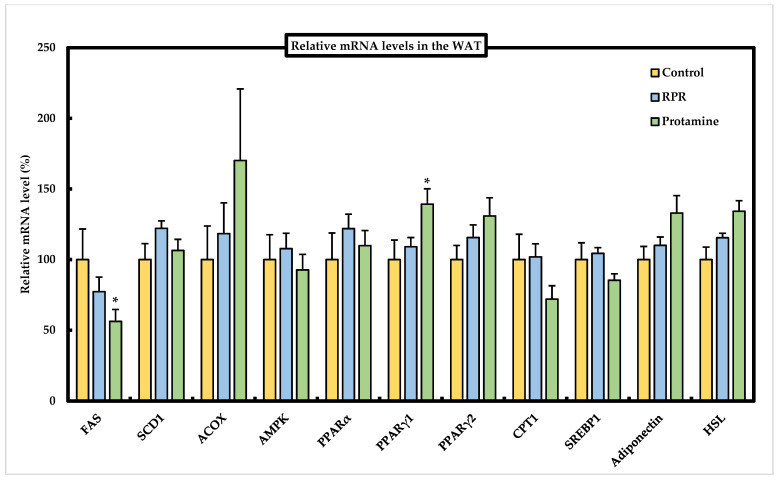
Effect of RPR or protamine on relative mRNA levels in the WAT of C57BL/6J mice with high-fat diet-induced obesity. The relative mRNA levels in WAT were measured after 50 days of high-fat diet treatments in mice. The mRNA levels of the various genes in the control group are considered to be 100%. Means ± SEM (*n* = 8). Differences from the control were calculated using Dunnett’s test (* *p* < 0.05). The relative mRNA levels were measured by real-time PCR.

**Figure 8 nutrients-13-02501-f008:**
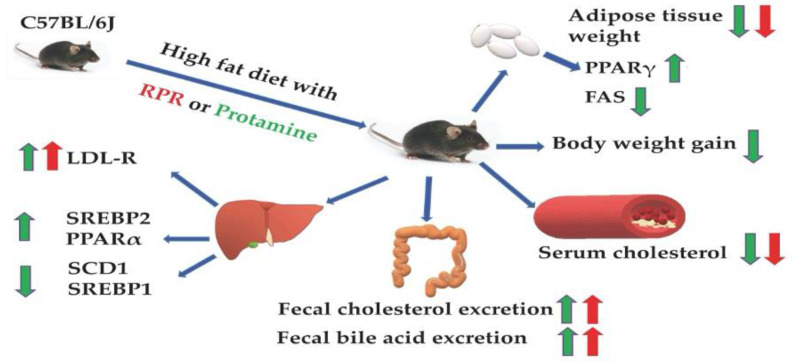
Mechanism of the anti-obesity and hypocholesterolemic actions of RPR and protamine. RPR and protamine ameliorate hypercholesterolemia through the increase in fecal steroid excretion and LDLR activation. The anti-obesity effects of protamine link to upregulation of adipocyte PPARγ1 mRNA and hepatic PPARα mRNA and downregulation of hepatic SCD1 mRNA via SREBP1 mRNA and adipocyte FAS mRNA. RPR derived from protamine functions a crucial role in the anti-obesity action of protamine by evaluating the effective dose of adipose tissue weight loss. The red arrows represent RPR’s effects, and the green arrows represent protamine’s effects.

**Table 1 nutrients-13-02501-t001:** Experimental groups of mice fed a high fat diet (HFD).

(1) control (Control group)	25% (*w*/*w*) casein	HFD	*n* = 8
(2) 0.5% RPR (RPR group)	24.5% (*w*/*w*) casein + 0.5% (*w*/*w*) RPR *	HFD	*n* = 8
(3) 5% protamine (Protamine group)	20% (*w*/*w*) casein + 5% (*w*/*w*) protamine **	HFD	*n* = 8

* RPR (Peptide institute, purity > 95%), ** protamine sulfate (Sigma, P4380).

**Table 2 nutrients-13-02501-t002:** Primer used for real-time PCR.

Gene	Forword Primers (5′➝3′)	Reverse Primers (5′➝3′)	NCBI Gen Bank
*ABCA1*	CTTCCCACATTTTTGCCTGG	AAGGTTCCGTCCTACCAAGTCC	NM_013454.3
*ACOX*	AGCGAGCCAGAGCCCCAG	TCAGGCAGCTCACTCAGG	NM_015729.3
*Adiponectin*	TACAACCAACCAACAGAATCATTATGACGG	GAAAGCCAGTAAATGTAGAGTCGTTGA	NM_009605.5
*AMPK*	CATGGCTGAGAAGCAGAAGCAC	CTTAACTGCCACTTTATGGCCTG	NM_178143.2
*CPT1*	GGATCTACAATTCCCCTCTGC	GCAAAATAGGTCTGCCGACA	NM_013495.2
*CYP7A1*	AGCAACTAAACAACCTGCCAGTACTA	GTCCGGATATTCAAGGATGCA	NM_007824.3
*FAS*	TTCCAAGACGAAAATGATGC	AATTGTGGGATCAGGAGAGC	NM_007988.3
*HMGCR*	CCTGGGCCCCACATTCA	GACATGGTGCCAACTCCAATC	NM_001360165.1
*HSL*	ACCGAGACAGGCCTCAGTGTG	GAATCGGCCACCGGTAAAGAG	NM_001039507.2
*LDLR*	TGTGAAAATGACTCAGACGAACAA	GGAGATGCACTTGCCATCCT	NM_001252658.1
*PPARα*	TCAGGGTACCACTACGGAGTTCA	CCGAATAGTTCGCCGAAAGA	NM_001113418.1
*PPARγ1*	GGTGAACCACTGATATTCAGGACA	TGTGTCAACCATGGTAATTTCAGT	NM_001127330.2
*PPARγ2*	TGAGCACTTCACAAGAAATTACC	TGCGAGTGGTCTTCCATCAC	NM_011146.3
*SCD1*	TTCTTGCGATACACTCTGGTGC	CGGGATTGAATGTTCTTGTCGT	NM_009127.4
*SREBP1*	CTTTGGCCTCGCTTTTCGG	TGGGTCCAATTAGAGCCATCTC	NM_001313979.1
*SREBP2*	GTGCGCTCTCGTTTTACTGAAGT	GTATAGAAGACGGCCTTCACCAA	NM_033218.1
*β-actin*	TGTCCACCTTCCAGCAGATGT	AGCTCAGTAACAGTCCGCCTAGA	NM_007393.5

## Data Availability

The data supporting the findings reported herein are available on request, from the corresponding author.

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
