# Peer review of "Anti-Obesity and Hypocholesterolemic Actions of Protamine-Derived Peptide RPR (Arg-Pro-Arg) and Protamine in High-Fat Diet-Induced C57BL/6J Mice"

_nutrients, 2021, doi:10.3390/nu13082501_

Round 1

Reviewer 1 Report

I carefully read the manuscripts by Maihemuti Mijiti et al., it is well-written and results are clearly explained. However, there are some major concerns regarding some parts of the manuscript.

My specific major comment follow below;

Comment 1: For studying the effect of protamine/RPR intake in fat absorption and in weight reduction,

Please describe in much more detail how to administrate HFD: protaimine, HFD: RPR such as intragastric administration using a bulbed needle. And, include dose of protamine (or RPR ) per kg of animal. Additionally, explain whether the amount of daily food intake was actually even in all animals of three groups. Please, add the table explaining the details of each group.

Comment 2: Have you ever measured plasma insulin and glucose level in animals? If so, represent those value in figure or table, otherwise this should be discussed in limitation.

Because the insulin has effects on both the lipogenesis and lipolysis and hypocholesterolemia can be related with the insulin level. In addition, the promotion of TG storage in fat is one of the most important of the actions of insulin.

Comment 3: [Results, page 6, Figure 4 (A) and (B)]

Please, why don't you insert the level of free fatty acid, HDL-cholesterol, and LDL-cholesterol in the Figure 4 (A) and (B)?

In order to clarify whether the visceral deposition of fat was reduced by inhibition of fat absorption through protamine intake, it is necessary to additionally show the levels of free fatty acids and HDL-cholesterol of dietary TG metabolites.

In particular, according to the results [Results, page 7, lines 333- 344], authors mentioned that the lipid metabolism-related genes-SREBP2, HMGCR, and LDLR expression levels were upregulated in the protamine/RPR group. Therefore, it is necessary to present the circulating LDL-cholesterol level.

Comment 4: [Discussion, page 10, lines 461-465]

You suggested that the decrease in weight gain in mice caused by protamine feeding is due to increasing energy expenditure and oxygen consumption due to enhanced fatty acid oxidation and thermogenesis in discussion. To prove this hypothesis, you need to measure the brown adipose tissues, or check the basal metabolic rate of animals as well.

As you know, Brown adipose tissue/adipocyte is the most important thermogenic functional unit, which tissue metabolizes nutrients such as fatty acids and glucose and converts the energy to thermo energy. Also, accumulating evidences suggest that brown adipose tissue function is inversely associated with BMI, which indicates that the activation of brown adipose tissue has potential translational implication in obesity. 

Author Response

[Our Reply for Reviewer 1 Report]

We are grateful for the comments from Reviewers on our manuscript, as these were very insightful and helped us to improve our manuscript.

We corrected many sentences with a red color in revised manuscript.

[Reviewer 1 Report]

Comments and Suggestions for Authors

I carefully read the manuscripts by Maihemuti Mijiti et al., it is well-written and results are clearly explained. However, there are some major concerns regarding some parts of the manuscript.

My specific major comment follow below;

Comment 1: For studying the effect of protamine/RPR intake in fat absorption and in weight reduction,

Please describe in much more detail how to administrate HFD: protaimine, HFD: RPR such as intragastric administration using a bulbed needle. And, include dose of protamine (or RPR) per kg of animal. Additionally, explain whether the amount of daily food intake was actually even in all animals of three groups. Please, add the table explaining the details of each group.

(Our reply 1) Thank you very much for your valuable comments. As reviewer’s suggestion, we added Table 1 in revised manuscript. There are three experimental groups as shown in Table 1. Each group was given free access to drinking water and fed the experimental diets for 50 days (P2, L79-90). The food intake was recorded daily during the feeding period and there was no significant difference in the food intake on

each day among the groups. The average daily intake of protamine or RPR is 4.71±0.17 g/kg body weight or 0.5±0.02 g/kg body weight respectively for 50 days. These explanations are included in revised manuscript (P4, L160-163).

Comment 2: Have you ever measured plasma insulin and glucose level in animals? If so, represent those value in figure or table, otherwise this should be discussed in limitation. Because the insulin has effects on both the lipogenesis and lipolysis and hypocholesterolemia can be related with the insulin level. In addition, the promotion of TG storage in fat is one of the most important of the actions of insulin.

 (Our reply 2) As you pointed out above comments of insulin and glucose, they are important factor to discuss lipogenesis, lipolysis and hypocholesterolemia. Thus, we will measure the plasma insulin and glucose levels in mice fed RPR or protamine in future study. We rewrote discussion in our revise manuscript. (P9, L406-409)

Comment 3: [Results, page 6, Figure 4 (A) and (B)]

Please, why don't you insert the level of free fatty acid, HDL-cholesterol, and LDL-cholesterol in the Figure 4 (A) and (B)?

In order to clarify whether the visceral deposition of fat was reduced by inhibition of fat absorption through protamine intake, it is necessary to additionally show the levels of free fatty acids and HDL-cholesterol of dietary TG metabolites.

In particular, according to the results [Results, page 7, lines 333- 344], authors mentioned that the lipid metabolism-related genes-SREBP2, HMGCR, and LDLR expression levels were upregulated in the protamine/RPR group. Therefore, it is necessary to present the circulating LDL-cholesterol level.

 (Our reply 3) As reviewer’s suggestion, we conducted additional experiments to measure HDL-cholesterol and non-HDL-cholesterol. We added these results in revised manuscript (P6, L211-214, Fig. 4 a, b). Serum HDL-cholesterol level was significantly decreased in the protamine group or RPR group, and non-HDL-cholesterol was significantly decreased in the protamine group (Fig. 4 a). It has been reported that total cholesterol, HDL-cholesterol and LDL-cholesterol was significantly decreased in rats fed a protamine for 7 weeks (Takahashi, Y. et al. Fish Sci 77, 1045–1052 (2011)). These results are similar to our study. Also, we will investigate the level of free fatty acid to clarify the protamine or RPR effect in future study. We rewrote discussion in revised manuscript (P9, L401-406).

Comment 4: [Discussion, page 10, lines 461-465]

You suggested that the decrease in weight gain in mice caused by protamine feeding is due to increasing energy expenditure and oxygen consumption due to enhanced fatty acid oxidation and thermogenesis in discussion. To prove this hypothesis, you need to measure the brown adipose tissues, or check the basal metabolic rate of animals as well.

As you know, Brown adipose tissue/adipocyte is the most important thermogenic functional unit, which tissue metabolizes nutrients such as fatty acids and glucose and converts the energy to thermo energy. Also, accumulating evidences suggest that brown adipose tissue function is inversely associated with BMI, which indicates that the activation of brown adipose tissue has potential translational implication in obesity. 

 (Our reply 4) As reviewer’s comments, brown adipose tissue is the most important tissue for thermogenesis. To confirm our hypothesis, we will measure the brown adipose tissues, or check the basal metabolic rate of animals in future study. These explanations are included in revised manuscript (P10, L446-448).

Reviewer 2 Report

Dear Authors,

The manuscript describes an animal study conducted to estimate the influence of protamine and RPR administration on the features of obesity in C57BL/6J mice. The authors concluded that RPR and protamine could "ameliorate disorders related to obesity and hypercholesterolemia and  RPR derived from protamine plays a crucial role in the anti-obesity action of protamine"

I have numerous considerations regarding missing elements in the manuscript.

  1. The names of studied subgroups introduced by Authors, such as 5% protamine or 0.5% PRP, make results difficult to read. Percent values information should restrict to the material and methods section. Please, consider renaming subgroups.
  2. The last part of the introduction section should be more consolidated. For example, the description of protein Cutter usage and Fig1 should move to the material and methods section.
  3. Please explain briefly in the text the protamine action on TG level according to cited work.
  4. Please, provide the general amplification protocol for real-time PCR, particular annealing temperatures for each primer's pair. In addition, please provide a source of primers sequences (commercially available, from databases, from other studies, self-designed?)
  5. Please rewrite the ultimate conclusion of the work. The results from the study consider transcripts level only without any data about the protein level of studied genes. Therefore, it is pretty hard to conclude from transcript level only that the level of respective proteins will behave similarly.
  6. Please put the amount of RNA (ng) was reversely transcribed in the study
  7. In the section "Biochemical Analysis" please briefly describe applied biochemical analysis.
  8. Please extend 2.4. Statistical analyses with more detailed information (is a Dunnet a post hoc test?). Provide a statistical software name. Please unify the usage of SEM or SE in the text of the manuscript and Figs.
  9. How many housekeeping genes have the Authors applied as a control in the study? Only beta-actin?
  10. Please clarify the section about groups design in the experiment. How many groups of animals were in the study?

“HFD  control group, HFD with 0.5 % protamine-derived peptide (RPR) (Peptide institute, purity 104 > 95 %): 0.5 % RPR group and HFD with 5 % protamine sulfate (Sigma, P4380): 5 % protamine group. Was High-Fat diet applied for every group? Why are results provided for three groups only? Is the control in Fig2 an HFD control group? Please clarify descriptions.

  1. Is the control on the y axis the same as the control bar in the plot? Please clarify the information in Fig6.
  2. English language should be improved
  3. “The red arrows represent RPR’s effects” According to results, only LDLR mRNA level increased significantly compared to control after PRP treatment. Why in figure 8 are there several red arrows indicated changes in other transcripts? The figure should be corrected. Please notice that protamine and PRP actions are not equal while rewriting the conclusion according to the results. This fact should also be briefly discussed. The conclusion statement: “Particularly, RPR derived from protamine plays a crucial role in the anti-obesity action of protamine” is not supported by presented results and should be rewritten. Authors cannot conclude that PRP ameliorates diseases related to obesity because they did not evaluate such conditions in their study.

Author Response

[Our Reply for Reviewer 2 Report]

We are grateful for the comments from Reviewers on our manuscript, as these were very insightful and helped us to improve our manuscript.

We corrected many sentences with a red color in revised manuscript.

[Reviewer 2 Report]

Comments and Suggestions for Authors

Dear Authors,

The manuscript describes an animal study conducted to estimate the influence of protamine and RPR administration on the features of obesity in C57BL/6J mice. The authors concluded that RPR and protamine could "ameliorate disorders related to obesity and hypercholesterolemia and RPR derived from protamine plays a crucial role in the anti-obesity action of protamine"

  1. The names of studied subgroups introduced by Authors, such as 5% protamine or 0.5% PRP, make results difficult to read. Percent values information should restrict to the material and methods section. Please, consider renaming subgroups.

(Our reply 1) Thank you very much for your valuable comments. As reviewer’s suggestion, we changed the name of studied subgroups from 5 % protamine to protamine and 0.5 % RPR to RPR and included percent values information only in Materials and Methods section. These changes are involved in revised manuscript (Table 1). These changes in subgroup’s name are applied through the manuscript.

  1. The last part of the introduction section should be more consolidated. For example, the description of protein Cutter usage and Fig1 should move to the material and methods section.

(Our reply 2) According to reviewer’s suggestion, we revised the last part of the introduction section and the description of PeptideCutter usage moved to the Materials and Methods (P2, L68-73). Also, Fig.1 moved to the Results (P4, Fig.1). The results of PeptideCutter was added in revised manuscript (P4, L131-136).

  1. Please explain briefly in the text the protamine action on TG level according to cited work.

(Our reply 3) As reviewer’s comment, we cited work regarding the effects of protamine on fecal TG levels and explained it in revised manuscript. (P2, L50-52)

  1. Please, provide the general amplification protocol for real-time PCR, particular annealing temperatures for each primer's pair. In addition, please provide a source of primers sequences (commercially available, from databases, from other studies, self-designed?

(Our reply 4) According to reviewer’s suggestion, we have added the information of real-time PCR. The amplification protocol as follows; the temperature profile for the reaction was an initial denaturation stage of 95 °C at 30 sec, then a two-step program was developed for 40 cycles including denaturation at 95 °C for 5 sec, an annealing step and extension step at 60 °C for 30 sec, respectively. The following primers were used in the assay based on the NCBI databases (Table 2). We have added these explanations and a source of primers sequence in Materials and Methods (P3, L117-121, Table 2). To make it easier to understand, we added Table 2 for primers (P3).

  1. Please rewrite the ultimate conclusion of the work. The results from the study consider transcripts level only without any data about the protein level of studied genes. Therefore, it is pretty hard to conclude from transcript level only that the level of respective proteins will behave similarly.

(Our reply 5) As reviewer’s comments, our description did not clearly distinguish between transcripts and protein in original manuscript. Thus, we revised our Conclusions in revised manuscript (P11, L492-500, P12, L544-546).

  1. Please put the amount of RNA (ng) was reversely transcribed in the study

(Our reply 6) We used 5500 ng of RNA for the reverse transcription. We added this explanation to Materials and Methods in revised manuscript (P3, L113).

  1. In the section "Biochemical Analysis" please briefly describe applied biochemical analysis.

(Our reply 7) We conducted biochemical analysis as follows; various lipid concentrations were determined using commercially available kits as follows: plasma, liver, and fecal triglyceride with Triglyceride E-test Wako (Wako Pure Chemical, Osaka, Japan) and plasma, liver, and fecal cholesterol with Cholesterol E-test Wako (Wako Pure Chemical, Osaka, Japan), serum HDL-cholesterol with HDL-Cholesterol E-test Wako (Wako Pure Chemical, Osaka, Japan), liver, and fecal phospholipid with Phospholipid C-test Wako (Wako Pure Chemical, Osaka, Japan), and fecal bile acid with TBA-test Wako (Wako Pure Chemical, Osaka, Japan). Liver and fecal lipids were extracted by the method of Folch et al. (J. Biol. Chem. 226, 497-509 (1957)), and total lipids were determined gravimetrically by the method of Nagaoka et al (J. Nutr. 129, 1725-1730 (1999)). As reviewer’s comments, we added these explanations to Biochemical analyses(P3, L102-110).

  1. Please extend 2.4. Statistical analyses with more detailed information (is a Dunnet a post hoc test?). Provide a statistical software name. Please unify the usage of SEM or SE in the text of the manuscript and Figs.

(Our reply 8) Dunnet’s test is a post hoc test. The statistical significance of differences was evaluated by one-way ANOVA and Dunnett’s tests. The statistical analysis was performed by JSTAT software. This software has actually been used in other study. (Miyashiro, S.; Yamada, Y.; Muta, T.; ishikawa, H.; Abe, T.; Hori, M.; Oka, K.; Koshikawa, F.; Ito, E., Activation of the orbitofrontal cortex by both meditation and exercise: a near-infrared spectroscopy study. PLoS One 2021, 16, e0247685).

As reviewer’s comments, we added more detailed information to statistical analyses and unified the usage of SEM in revised manuscript (P4, L126-128, Fig2-7).

  1. How many housekeeping genes have the Authors applied as a control in the study? Only beta-actin?

(Our reply 9) We used only β-actin. As there are no significant differences in β-actin mRNA levels between the groups in this experiment, we used β-actin as a housekeeping gene.

  1. Please clarify the section about groups design in the experiment. How many groups of animals were in the study?

“HFD control group, HFD with 0.5 % protamine-derived peptide (RPR) (Peptide institute, purity 104 > 95 %): 0.5 % RPR group and HFD with 5 % protamine sulfate (Sigma, P4380): 5 % protamine group. Was High-Fat diet applied for every group? Why are results provided for three groups only? Is the control in Fig2 an HFD control group? Please clarify descriptions.

(Our reply 10) In this study, the experimental group was three groups: HFD control group, RPR group, and protamine group. The high-fat diet was applied for every group and protamine or RPR was added to high-fat diet. To make it clearer, we added Table 1 and rewrote the description in revised manuscript (P2, L83, Table1).

  1. Is the control on the y axis the same as the control bar in the plot? Please clarify the information in Fig6.

(Our reply 11) The control on the y axis is the same as the control bar. The mRNA levels of the various genes in the control group are considered to be 100 %. However, the y axis notation can be confusing. We changed the y axis notation from % of control to relative mRNA level (%) and added these explanations in revised manuscript (P8, L341-342, P9, L384-385, Fig.6, Fig.7).

  1. English language should be improved

(Our reply 12) Thank you very much for your valuable comments.

We have extensively reviewed English language of our manuscript.

  1. “The red arrows represent RPR’s effects” According to results, only LDLR mRNA level increased significantly compared to control after PRP treatment. Why in figure 8 are there several red arrows indicated changes in other transcripts? The figure should be corrected. Please notice that protamine and PRP actions are not equal while rewriting the conclusion according to the results. This fact should also be briefly discussed. The conclusion statement: “Particularly, RPR derived from protamine plays a crucial role in the anti-obesity action of protamine” is not supported by presented results and should be rewritten. Authors cannot conclude that PRP ameliorates diseases related to obesity because they did not evaluate such conditions in their study.

(Our reply 13)

As reviewer’s suggestion, we deleted the red arrow which shows the effect of RPR on the transcript of lipid metabolism-related genes in Figure 8 in revised manuscript (Fig. 8). Based on the above, we added to the explanation regarding the difference on the physiological action between protamine and RPR in Conclusions in revised manuscript (P11, L492-500). Also, as reviewer’s comment, we rewrote conclusion statement in Conclusions in revised manuscript (P1, L22-28, P11, L492-502, P12, L544-547).

Round 2

Reviewer 1 Report

The authors have made a commendable effort to revise the manuscript according to the reviewers' suggestions and the manuscript has improved substantially.